# Effect of Micro-Scale Er on the Microstructure and Fluidity of ZL205A Alloy

**DOI:** 10.3390/ma12101688

**Published:** 2019-05-24

**Authors:** Tingbiao Guo, Bing Wang, Zhanfei Zhang, Quanzhen Sun, Yuhua Jin, Wanwu Ding

**Affiliations:** 1School of Materials Science and Engineering, Lanzhou University of Technology, Lanzhou 730050, China; 15002531091@163.com (B.W.); ztswbing@163.com (Z.Z.); wsr940808@163.com (Q.S.); yhjin8686@163.com (Y.J.); dingww@lut.cn (W.D.); 2State Key Laboratory of Gansu Advanced Non–ferrous Metal Materials, Lanzhou University of Technology, Lanzhou 730050, China

**Keywords:** ZL205A, rare earth Er, fluidity, microstructure, T5 heat treatment

## Abstract

The effect of Er addition on the fluidity and microstructure transformation of the as-cast and T5 heat-treated ZL205A alloys was investigated by optical microscope (OM), scanning electron microscopy (SEM), X-ray diffraction (XRD) and energy dispersive spectrometer (EDS). The fluidity of the liquid metal after adding Er was tested and the fracture characteristics of the material were analyzed. The results indicated that Er was mainly dissolved into an α–Al matrix near the grain boundaries (GBs). It is easily segregated and enriched in the intersection of the GBs or the interface between the α and θ phase, which caused the intermetallic compounds to be distributed along the GBs to the neck and to fuse. Er could also inhibit the diffusion of Cu atoms in the process of solid solution, so that increased the residual eutectic structures in the crystal, while accelerating the precipitation progress of the Guinier–Preston (GP) zone and θ’ phase and increasing precipitation of the θ phase. A small amount of precipitation of θ phase and micro-scale Er (0.1–0.5 wt %) can significantly increase the fluidity and reduce the casting defects, which can effectively improve the castability of the ZL205A alloy. The interface between the (Al_8_Cu_4_Er) phase and matrix is the main area of microcracks, through analyzing the fracture morphology.

## 1. Introduction

Aluminum alloy has attracted more and more attention as a kind of dual-use strategy material, due to the good combined properties, such as high strength and toughness, low density, excellent process ability, and good corrosion resistance [1]. Casting Al–Cu–Mn alloys are widely used in aviation, aerospace, automotive, machinery, and other industrial applications [2]. Copper is used as a precipitation strengthening agent in aluminum alloy [3]. The addition of 5.0 wt % Cu can lead to the alloy’s excellent strength and toughness when subjected to natural and artificial aging [4,5]. Mn addition up to 0.4 wt % forms T (Al_12_CuMn_2_) phase with Cu and Al, which is dispersed and strengthened when subjected to solution treatment [5]. Al–Cu–Mn enjoys high specific intensity, excellent processing performance, and good corrosion resistance [6], which is a very important alloy system in material application. Nonetheless, casting defects form easily under the normal casting conditions, due to a wide range of crystallizing temperature and high hot cracking sensibility, which limits the further application of the alloys in high-performance structure fields [7].

Recent studies show that the addition of rare earth (RE) micro-alloy is an effective method to improve the casting properties of aluminum alloy [8,9,10]. Researchers have demonstrated that RE elements as a grain refiner or recrystallization inhibitor [11,12] can improve the castability of aluminum alloy by reducing the hot cracking tendency [13], and form coherent L12-ordered precipitates, such as Al_3_Sc, Al_3_Er, and Al_3_Zr, which can effectively improve the yield strength, fatigue strength, thermal stability, and creep resistance of aluminum alloys [14,15,16]. With the many micro-alloying elements, Sc has a significant effect, but is difficult to use, widely because of its high cost [14]. Instead, much attention was focused on rare earth Erbium (Er), which is cheaper than Sc. Er can form the coherent or semi-coherent Al_3_Er particle lattice with aluminum matrix in Al–Zn–Mg, Al–Mg, and other aluminum alloys [17,18]. The phase owns the cubic system belonging to the Pm3m space group, like Al_3_Sc and Al_3_Zr. The Al_3_Er phase has the characteristics of high melting point and good stability, which can improve the strength of the alloy. Therefore, it has attracted the attention of many researchers [19,20]. Furthermore, Er can improve the yield strength and recrystallization resistance of pure aluminum [21], and increase the thermal stability of Al–Mg alloy [21]. The addition of Er can form the precipitation of Al_3_Er phase and cause remarkable grain refinement so that these may improve the hardening effect of Al–Zn–Mg alloy [22]. Furthermore, after heat treatment, the addition of Er could restrain the precipitation of Ω phase and promote the formation of θ’ phase, which could refine the dendritic substructure to improve the mechanical properties of as-cast Al–Cu–Mg–Ag alloy [23]. Thus, the addition of Er to improve alloy properties has been the focus of many studies.

In this study, micro-scale Er was added to the ZL205A alloy to analyze the principle of phase transition and the microstructure evolution of the as-cast and T5 heat-treated alloys. Moreover, the effect of the Er content on the fluidity and fracture morphology was further investigated.

## 2. Experimental Procedure

ZL205A, fabricated by Beijing Aeronautical Materials Research Institute (Beijing, China), was selected as the master alloy, and the micro-scale Er was added into the ZL205A alloy using Al–Er intermediate alloy to achieve micro-alloying. Table 1 lists the chemical composition of the alloy studied. The amounts of Al–Er and Al–Cu intermediate alloys are shown in Table 2. The ZL205A master alloy was weighed as 3 kg per furnace, then was prepared by crucible resistance furnace. When the aluminum ingot and the Al–Cu master alloy were completely melted at 730 °C, the 0.4 wt % C_2_Cl_6_ was added to be refined in the bell jar. The slag was skimmed after heat preservation and stood for about 10 min. The temperature was then raised to 750 °C, and the corresponding intermediate alloys were added and stirred uniformly. The sample was held for 10 min after addition of the Al–Cu and Al–Er intermediate alloys. Finally, flowable spiral samples and microstructure samples were poured when the temperature dropped to 730 °C.

Samples of each component were divided into two groups: One group for T5 heat treatment (540 °C, 12 h solution treatment and room temperature water quenching; 150 °C, 10 h artificial aging treatment), and the other group for as-cast alloys. The cylindrical specimens with a height of 50 mm were cut from the center of microstructure samples, polished, and etched with 99.5% H_2_O + 0.5% HF to obtain metallographic specimens. The microstructure was observed by a MEF-3 optical microscope (OM, Leica, Germany), and the tensile fracture surface was observed by a JSM-6700F scanning electron microscope (SEM, Shimadzu Coporation, Kyoto, Japan). To identify the phase components of the samples, a JSM-5600Lv SEM with the energy dispersive spectrometer (EDS, Shimadzu Coporation, Kyoto, Japan) was performed on polished specimens. The phase of as-cast and T5 heat treatment samples were identified by D8 Advance X-ray diffraction (XRD, LYNXEYE dector, radius of goniometer 250 mm, the tube has an accelerating voltage of 40 kV, an emission current of 40 mA, CuKα, λ = 1.54156 Å, scanning speed of 10°/min, step size of 0.02°, angle from 2 Theta 20° and 100°).

## 3. Results and Discussion

### 3.1. The Microstructure of As-Cast ZL205A Alloy

Figure 1 shows the microstructure of as-cast alloy with Er contents of 0, 0.1, 0.3, 0.5 wt %. It can be seen that the dendritic structure of the alloy changes significantly with the different additions of Er. The grains of the as-cast structure are almost large-size equiaxed crystal and show serious segregation of precipitated phases at grain boundaries (GBs) (as shown in Figure 1a). When Er content is 0.1 wt %, the distribution of grains and precipitates become uniform, and the grain size is larger than the original structure. When the addition of Er is 0.3 wt %, the as-cast alloys have refined grain sizes of 40 μm, and the overall shape presents a loose flocculent form (as shown in Figure 1b). When Er exceeds 0.5 wt %, the grain shape changes from granular to flaky with increasing size. Some grains become coarser, of more than 60 μm. The grain shows uneven distribution, and small-size grains are mostly distributed among adjacent large grains.

Figure 2a–d are the SEM images of the as-cast alloy with different contents of Er. Figure 2 indicates that when the Er was added, the microstructure of alloy had varying degrees of refinement. In the original state of alloy, white phase particles that precipitated in the GBs had uneven distribution (as shown in Figure 2a). When the addition of Er is 0.1 wt %, the distribution of white second phase is more uniform than the original structure, and precipitation looks like a long strip. When the addition content of Er is 0.3 wt %, the size of white phase particles (θ phase, T phase) increases, and the amount of precipitation increases too. Grain sizes were refined. As shown in Figure 2d, when addition content of Er is 0.5 wt %, the branching of white phase structure at GBs is serious, and segregation and agglomeration structures exist at the junction of GBs. Meanwhile, grain sizes of the alloy were clearly increased. This is consistent with the change of alloy OM microstructure. It is shown that the improvement of alloy microstructure is very strong after adding 0.3 wt % Er.

To analyze the effect of Er content on the grain sizes of the as-cast alloys, 50 grains were selected randomly in the SEM (50 μm) images, then measured and calculated by Nano Measurer v1.2.5 software (Medium imitation, China). Figure 3 shows different grain sizes (max, min, mean, and median). With the increasing of Er from 0.1–0.5 wt %, the mean and median grain sizes varied only slightly with a center near 40 μm. The vertical lines showed that the size interval would gradually expand with the addition of increasing Er amounts. Specifically, the grain sizes are mainly 30 μm and 58 μm for the original alloy. When Er content increased to 0.5 wt %, the range was about 30 μm to 65 μm, which is clearly greater than the other samples.

The as-cast alloy containing 0.3 wt % Er was studied by SEM and EDS to analyze the microstructure and distribution of solute elements (as shown in Figure 4). Figure 4a shows the eutectic phase is about 5–7 μm along the GBs in the alloy (such as B-point morphology). In addition, second-phase particles, which precipitated in the GBs, were distributed in a grid-like manner in local. The results of EDS show that the Cu content is much higher than Mn at point B, where the Mn and Zr content is 0.3 wt % and 0.2 wt %, but it does not contain Er. Thus, the phase is θ-Al_2_Cu, containing Mn and Zr. At the same time, the phase is also θ-Al_2_Cu, containing Mn and Zr at point C. It is seen that the as-cast alloy structure is mainly composed of α-Al and θ-Al_2_Cu, precipitated by eutectic reaction (as shown by the white shape in Figure 4a). With the irregular network and long strip, the θ-Al_2_Cu is distributed at the GBs. With the granular material, some of the θ-Al_2_Cu alloy existed inside the grains. From the SEM topography and EDS analysis, it is known that Er has a significant segregation phenomenon in the alloy.

Figure 5 shows the elemental distribution of the as-cast alloy. The density of Cu in GBs is significantly greater than the inside of the grain (as shown in Figure 5a,b). Er is mainly dissolved into the matrix phase near the GBs, and is enriched at the junction between branches and turning points, causing the white phase to be distributed along the necks of GBs and fuse into short rods and granular material. Furthermore, partial white phases were recombined as block shapes in the boundary regions where Er content was low. Adding Er can generate massive shrinkage porosities and pinholes, which dramatically reduce the relative density in the as-cast alloy (as shown in Figure 5a).

In this work, it can be concluded that the effect of Er, refining in the local enrichment zone between adjacent large grains, is not obvious on overall average grain size refinement of ZL205A alloy. It can be attributed to the following points: First of all, in the solidification process of castings, Er is difficult to be squeezed into the front of a solid–liquid interface to increase the undercooling of components, because the cooling rate is fast and the atomic radius of element Er (0.245 nm) is close to Al (0.143 nm). Therefore, its refinement effect is not obvious, but there are small grains in the local micro-area. Secondly, because the Er element is located in Lanthanide 11 and the arrangement of extranuclear electrons is 4f^12^6s^2^, it is easy to lose two “s” electrons in the outermost layer and one electron in the 4f layer to become a trivalent cation. So, it is high chemical activity. Er tends to be distributed in the tensile strain region, near the phase boundary and GBs. It easily reduces the surface tension and interfacial energy of the two-phase interface, which leads to the segregation near the GBs and the crystal surface. Therefore, the growth of some grains was restrained, while some grains’ size increased. Combined with the EDS results, it was found that Er elements were not obviously detected in the field of view, indicating that Er elements did not exist in the form of dispersive particles. Er elements might be segregated at the intersection of GBs, resulting in weaker grain refinement. Furthermore, Er elements were not detected by EDS at the GBs. A large number of Er elements, which did not exist in the form of dispersed particles, were segregated at the intersection of GBs. The Al–Cu–Er ternary phase diagrams of the Al-rich region [24,25,26] suggest that the Al_3_Er and Al_8_Cu_4_Er eutectic phases may precipitate when the Er content reaches or exceeds the eutectic composition (shown in Figure 6). In this experiment, it is detected by the X-ray diffraction (XRD) that when the content of Er is high in the segregation zone near the GBs, the eutectic composition can be obtained, and the (Al_8_Cu_4_Er) eutectic phase is formed (shown in Figure 7).

In addition, a large number of shrinkage and pinholes are produced after adding Er, which reduces the relative density of as-cast alloys. The void site is filled by the low melting point phase when the alloy forms a dendritic network skeleton in the solidification process, and the GBs’ network structure of the alloy is developed, which runs through the entire field of view. The number of small volumes of liquid phases separated between adjacent dendrites increases, which generates dispersed shrinkage and porosity from the insufficient supply in the later stage of solidification. Moreover, due to the wide crystallizing range, there is a long-term solid–liquid coexistence of Al–Cu alloy. As a result, it easily formed micro-cracks at the GBs’ juncture during cooling shrinkage, which leads to a loose as-cast structure. The weakened intergranular cohesion strength results in adverse effects on the mechanical properties of the alloy.

### 3.2. Effect of Er on T5 Heat-Treated Microstructure

Figure 8a–d shows the SEM images of T5 heat-treated alloys with Er contents of 0, 0.1, 0.3, 0.5 wt %, respectively. Figure 8e–h shows the results of the corresponding EDS and XRD. The microstructures of the alloys after T5 heat-treatment were almost evolved into uniform equiaxed grains (as shown in Figure 8a–d). The entire composition was homogenized, and the relative density increased. It indicates that T5 heat-treatment can eliminate most of the micro segregation, and convert the residual phases, which exhibit dendritic shapes along the GBs of the as-cast alloys into large discontinuous granular forms to release the casting stress and distortion energy to a certain degree. As shown in Figure 8b, when the addition of Er is 0.2 wt %, the shape and size of grains are inconsistent, while the grain’s arrangement is relatively tight. The number of θ phases at the GBs is reduced and the generation of the small-sized grains is observed. When the Er content is 0.3 to 0.5 wt %, the width of GBs is decreased. At the same time, the morphology of white phases changed into particles, which segregated in the matrix, and the residual amounts of white phase concurrently increased. These residual phases usually present three models of dispersive particles—slender strips, short rods, or bulk shapes. In order to further investigate the effect of the addition of Er on the microstructure of the alloy, EDS analysis was carried out by adding 0.3 wt % and 0.5 wt % Er. The EDS results show that the Al_2_Cu is the main residual phase in this area. Some areas, labeled by the arrow, had a high Er concentration (shown in Figure 8c,d). Compared with the points e and f (shown in Figure 8c), it was found that the Er element was not detected in the white particle phase near the GBs, and an Er-rich region was formed at the end of the intersection of GBs. The existence of the Er-rich region increases the undercooling degree, which reduced the critical nucleation work so that the number of grains could increase. However, the Er-rich region is pinned at the end of the GBs, due to the slow diffusion. This phenomenon can lead to hindering the movement of dislocations and the migration of the sub-boundary, inhibit the extension and growth of the GBs, and reduce the connectivity and uniformity of the GBs. The content of Cu, Er, and Zr at point e is 21.68 wt %, 17.53 wt %, 0.3 wt %, respectively. The atomic ratio of Cu/Er is more than 4:1, so the phase should be Al–Cu phase or Al_8_Cu_4_Er phase, containing Er and Zr at GBs [27,28]. The existence of Al_8_Cu_4_Er was also confirmed by XRD (show in Figure 8g). The area in Figure 8d consisted of 46.97 wt % Al, 17.9 wt % Cu, 9.22 wt % Er, and 0.3 wt % Zr. The atomic ratio of Cu/Er is also more than 4:1. So this region is composed of the θ phase or the Al_8_Cu_4_Er phase containing Er and Cr [27,28]. It is noted that there were some carbon and oxygen impurities introduced through using a graphite crucible in the smelting and pouring process.

It is considered that during solution treatment at 540 °C, the solute atom fully diffuses, and most of the micro-segregation was eliminated. The EDS results (the gray phase, the white phase, the matrix phase) indicate that the concentration difference between Al and Cu becomes smaller than the as-cast alloy, and the structure of as-cast alloy is greatly improved. The added trace of Er diffuses to the GBs, which forms the Al_8_Cu_4_Er low melting eutectic phase instead of Al_3_Er phase. The Al_8_Cu_4_Er is an intermetallic compound with a low melting point and exists in a flaky form. The lamellar Al_8_Cu_4_Er phase is not only low strength, but also effectively consumes Cu atoms in the vicinity of the GBs to reduce the concentration Cu atoms in the supersaturation solid solution. As a result, the microscopic dispersed phase precipitated during aging is reduced.

### 3.3. Thermodynamic Analysis

Figure 9 shows the CCT and TTT diagrams of Al–5.0Cu–0.4Mn–0.2Ti–0.2Zr–0.25V–0.05B (wt %) alloy, simulated by JmatPro 10.0 thermo dynamic software (Medium imitation, China). The CCT diagram shows that the θ phase will precipitate first at 500 °C because the actual cooling speed is less than 1000 °C/h (right side of the cooling curve in Figure 9). The θ′ phase and the GP zone precipitated subsequently at 400 °C and 200 °C, respectively. However, when the alloy is cooled in air, it is limited by the diffusion kinetic conditions, and the total amount of precipitation is still small. Based on the TTT diagram, the solid–phase transformation of the Al–5.0Cu–0.4Mn–0.2Ti–0.2Zr–0.25V–0.05B (wt %) alloy occurred at 150 °C × 10 h artificial aging. The alloy generates solid transformation. Firstly, a large amount of GP zones are deposited, and then a small amount of θ’ phase is formed around 0.5 h. Finally, the alloy phase is mainly GP zone and θ′ phase, and there is little θ equilibrium phase. The formation of a GP zone and θ′ phase mainly depends on the diffusion of massive surplus vacancies and supersaturated solid atoms during the solid solution process.

As a surface-active element, Er is easily adsorbed in the interface of atomic groups or even in the cores, which reduces the interfacial energy and activation energy of nucleation during the solidification process and shortens the preparation time in the incubation period. These phenomena accelerate the precipitation progress of the GP zone and the θ’ phase, offsetting the TTT curves to the left side, which will precipitate the θ phase. These GP zones precipitate and the θ’ phase keep the crystallographic orientation partially coherent. The matrix usually exists as a form of lamellar and strip shapes. However, after T5 heat treatment, the precipitated phase is still dominated by the GP zone. The morphology of these precipitates is generally lamellar or strip-like, and the crystallographic relationship of the coherent phase with the matrix is partially coherent. In the GP zone, compared with the Al atom, the Cu atom radius is smaller, and the concentration of Cu atom is high, which will make the matrix beside the GP zone produce a large lattice distortion and stress field. These observations indicate that the phenomenon will hinder the movement of slipping dislocation and the sub-boundary to enhance the strength and hardness of the alloy. However, the presence of the lamellar or strip form had a severe effect on the fracture properties when the alloy was subjected to external force. The stress at the edge portion would preferentially exceed the yield limit and turn into a plastic deformation zone to form micropores and cracks.

### 3.4. Effect of Er on Fluidity

Figure 10 shows the fluidity and length curve of the different samples of the investigated alloy with different contents of Er. The addition of 0.1 wt % Er rapidly increases the fluidity length from 117 mm to 267 mm. When Er content was 0.3 wt %, the fluidity length decreased slightly, while still higher than the original length. Further increasing Er up to 0.5 wt %, the fluidity length increased to 400 mm, which reached the maximum length in this experiment. However, with the increases of Er, the surface smoothness of the samples reduced and some sand inclusion, such as shrinkage and other casting defects appeared. After all, micro-scale Er (0.1–0.5 wt %) can significantly increase the fluidity and reduce the casting defects, which can effectively improve the castability of the experimental alloy.

By analyses, the improvement of Er on the fluidity can attribute to the follow points: (1) Er element can shorten the crystallization temperature interval, reduce the solid–liquid two-phase residence time, and enhance the intergranular intercalation of as-cast microstructure in the late solidification stage; (2) it can absorb a large number of impurity elements, such as H and O, and reduce pinhole, stomata, and other casting defects (shown in Figure 10a); (3) furthermore, viscous resistance is an important factor to restrict the fluidity of the liquid alloy. The physical origin of the viscous resistance (or internal friction) is the momentum exchange. The attraction of the adjacent fluid layer occurs during the irregular molecular movement. When adding Er into the liquid ZL205A alloy, it can reduce the diffusion rates of Cu atoms. Thus, the mixing degree of the molecular transfer is reduced in the fluid layers, and the viscous resistance of the alloy is weakened. That aside, when the addition of Er is 0.3 wt %, the network structures with numerous branches are distributed along the GBs, which will be the obstacle for the fluidity of the liquid alloy. Thus, the length of the fluidity sample is shorter than the length of the content of 0.1 wt % and 0.5 wt % Er because the crystal bond strength is relatively weak. When Er increased to 0.5 wt %, the network structures decreased, and the length of fluidity sample continued to increase. The heterogeneous microstructure and impurity elements are inhaled during the pouring and solidification stages, which result in some casting defects (Figure 10a).

### 3.5. Fracture Performance

Figure 11a–c shows the tensile fracture morphologies of the as-cast ZL205A alloys with the Er addition of 0.1 wt %, 0.3 wt % and 0.5 wt %, respectively. Figure 11d–f represents the fracture surfaces of the corresponding component alloys after T5 heat-treatment. When the addition of Er is 0.1 wt %, the fracture mode is intergranular ductile fracture (as show in Figure 11a). This phenomenon can be attributed to some obvious tearing edges and dimples observed in the fracture morphology. When 0.3 wt % Er is added, the as-cast alloy shows a flocculent and loose structure, demonstrating a mixed mode of quasi cleavage and ductile fracture. As can be seen from Figure 11b,c when the addition of Er is 0.3–0.5 wt %, there are no obvious tearing edges in the field of view. However, with the increase of the Er content, the solid solubility was reduced both in the matrix and second phase to form a long strip cleft. Therefore, the structure becomes loose. Due to the cooling and diffusion rate, the intercrystalline structure is not close and expands into a long fracture during the stretching process. In addition, Figure 11a shows that there are some large-area cracks and the macroscopical shape of the as-cast alloy is loose and flocculent. After T5 treatment, with the increases of Er content, the toughness pits and micropores are greatly reduced, or become a shallow fossa morphology (as shown in Figure 11d–f). This observation indicates that the internal structure becomes dense, and the intercrystalline bond strength is increased. After all, the fracture mode gradually becomes brittle.

It was observed that the enhancement of ZL205A alloy is mainly due to the solution strengthening and dispersive precipitation strengthening of Cu atoms. The strengthening effect depends on the solid solubility, dispersion degree, and the existence form of the precipitate phase. The analysis results show that a part of Er reacted with Cu to form an Al_8_Cu_4_Er low-melting eutectic phase and the addition of Er did not have an obvious effect on solid solution strengthening. The formation of an Al_8_Cu_4_Er can not only weaken the strength of the alloy itself, but also effectively consume Cu atoms near the GBs to reduce its concentration in the supersaturated solid solution, which reduced the dispersoid precipitation during the aging process. With the increase of Er content, the eutectic structure, segregated along GBs, became granular and the degree of segregation agglomeration increased. The GBs network became a microporous structure, and the formation of inhomogeneous micropores near the inclusions in the matrix or at the junction of GBs, which could act as a crack source, is the main reason for the generation of the local stress and strain These micro-holes and cracks propagated and grew with each other under the action of external force, which eventually lead to the occurrence of fractures.

## 4. Conclusions

(1) Micro-scale Er (0.1–0.5 wt %) can significantly increase fluidity and reduce casting defects, which effectively improves the castability of the experimental alloy;

(2) Er is mainly a solid dissolved into the matrix phase near the GBs. It is easily segregated and enriched in the intersection of GBs or the interface between the α phase and θ phase, which causes the white phase that is distributed along the GBs to fuse. Er can also inhibit the diffusion of Cu atoms in the process of solid solution, increasing the residual eutectic structures in the crystal, while accelerating the precipitation progress of the GP zone and θ’ phase and increasing precipitation of the θ phase;

(3) The interface between the (Al_8_Cu_4_Er) phase and matrix is the main area of microcracks, seen through analyzing the fracture morphology.

## Figures and Tables

**Figure 1 materials-12-01688-f001:**
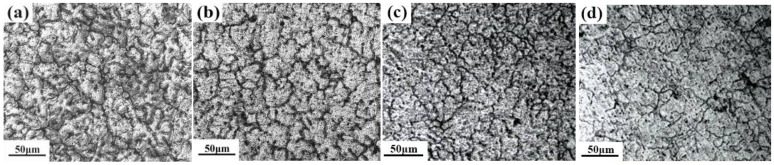
Microstructures of as-cast alloys with varying added content of Er: (**a**) 0%; (**b**) 0.1%; (**c**) 0.3%; (**d**) 0.5%.

**Figure 2 materials-12-01688-f002:**
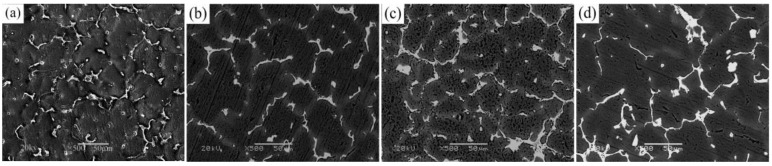
SEM images of as-cast alloys with varying added content of Er: (**a**) 0%; (**b**) 0.1%; (**c**) 0.3%; (**d**) 0.5%.

**Figure 3 materials-12-01688-f003:**
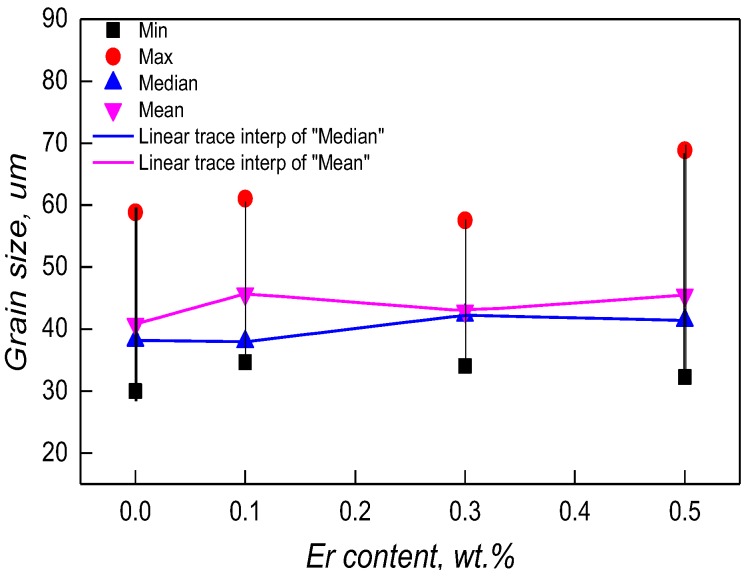
Change curves of grain size along with Er content.

**Figure 4 materials-12-01688-f004:**
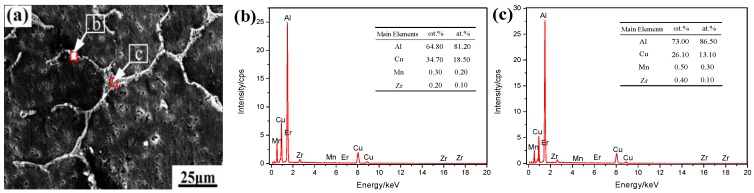
Scanning electron microscopy (SEM) and energy dispersive spectrometer (EDS) images of the as-cast alloy with addition of 0.3 wt % Er: (**a**) Morphologies; (**b**,**c**) are the EDS images of the as-cast alloy.

**Figure 5 materials-12-01688-f005:**
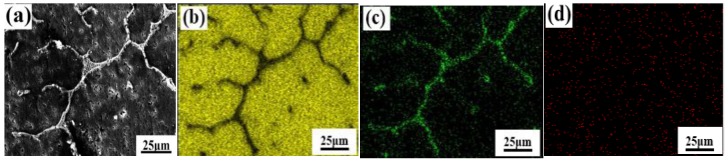
SEM plane scan analysis of 0.3 wt % Er under as-cast conditions: (**a**) Morphologies; (**b**) Al; (**c**) Cu; (**d**) Er.

**Figure 6 materials-12-01688-f006:**
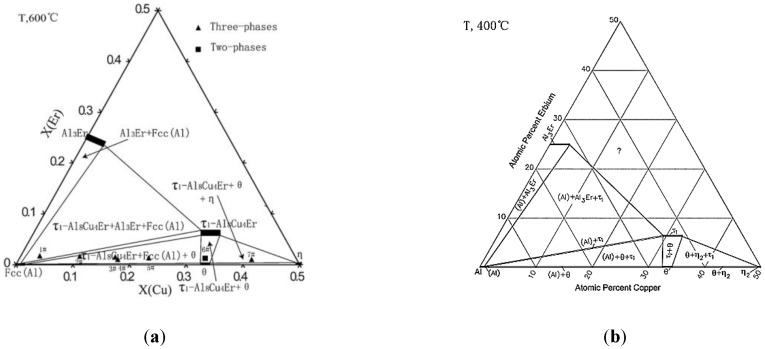
The Al–Cu–Er ternary phase diagrams at 600 °C and 400 °C [24,25,26].

**Figure 7 materials-12-01688-f007:**
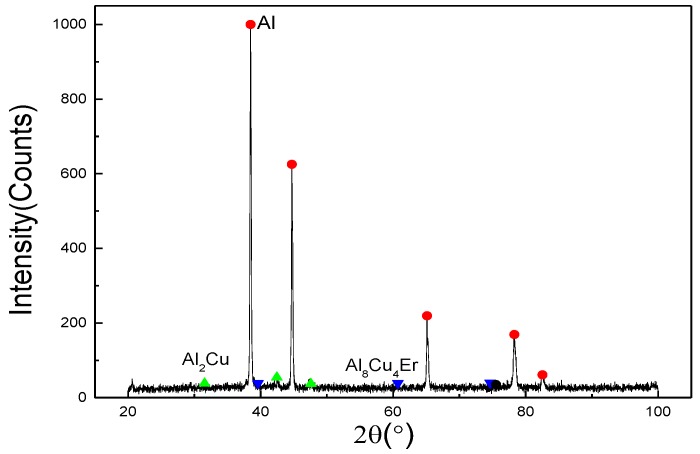
X-ray diffraction (XRD) spectrum of as-cast alloy after adding 0.3 wt % Er.

**Figure 8 materials-12-01688-f008:**
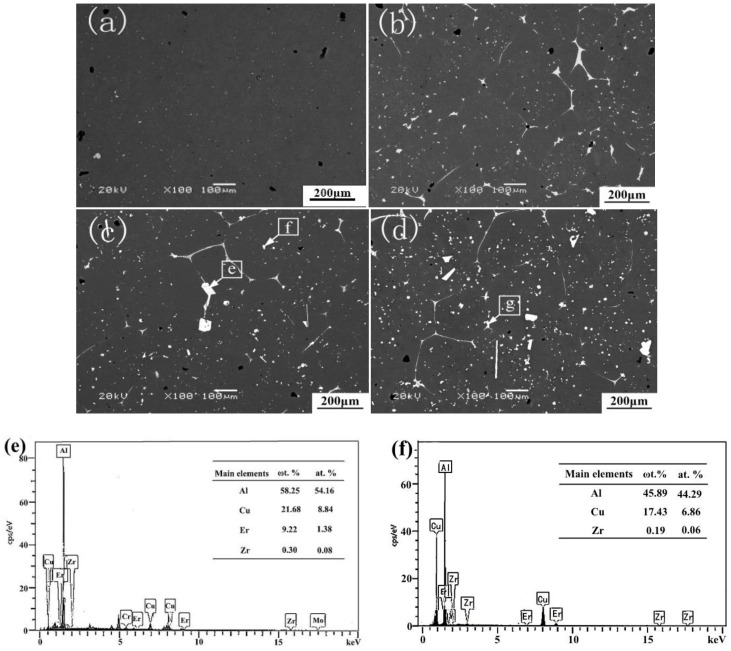
SEM images of T5 heat-treated alloys with different Er content: (**a**) 0; (**b**) 0.1 wt %; (**c**) 0.3 wt %; (**d**) 0.5 wt %; (**e**,**f**,**g**) EDS images of T5 heat-treated alloys with 0.3 wt % Er content; (**h**) XRD spectrum of T5 heat-treated alloys with 0.3 wt % Er content.

**Figure 9 materials-12-01688-f009:**
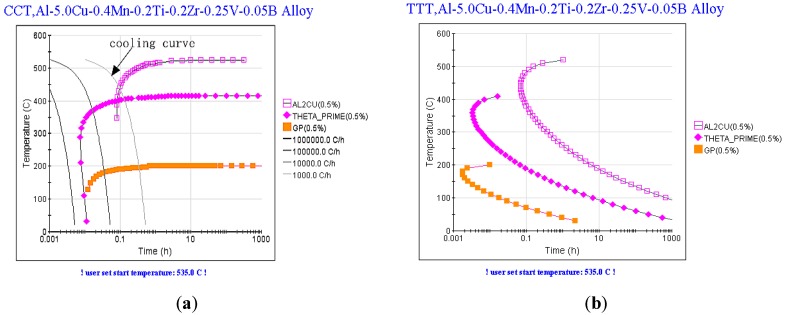
The CCT and TTT diagrams of Al–5.0Cu–0.4Mn–0.2Ti–0.2Zr–0.25V–0.05B (wt %) alloy simulated by JmatPro thermodynamic software: (**a**) CCT; (**b**) TTT.

**Figure 10 materials-12-01688-f010:**
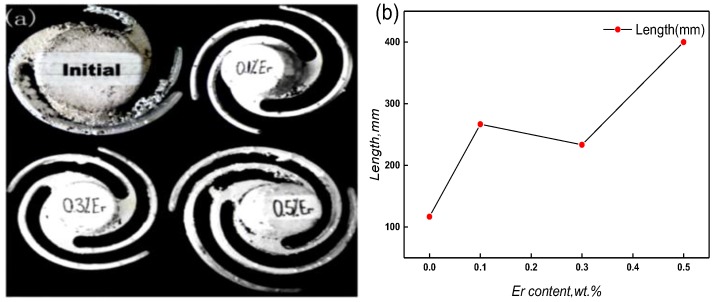
Fluidity samples (**a**) and length curve (**b**) with the addition of different Er content.

**Figure 11 materials-12-01688-f011:**
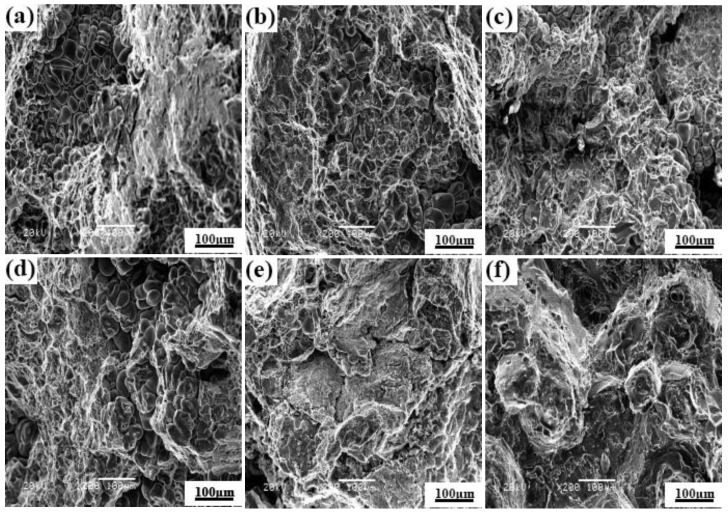
Fracture morphologies of the alloys with addition of different content of Er. As-cast: (**a**) 0.1 wt %; (**b**) 0.3 wt %; (**c**) 0.5 wt %. T5 heat treatment: (**d**) 0.1 wt %; (**e**) 0.3 wt %; (**f**) 0.5 wt %

**Table 1 materials-12-01688-t001:** Specific chemical compositions of the investigated alloy, wt %.

Cu	Mn	Ti	Cd	V	Zr	B	Al
5	0.4	0.2	0.2	0.25	0.2	0.05	Bal.

**Table 2 materials-12-01688-t002:** Furnace No. and addition amount of Al–Er intermediate alloy.

Furnace No.	1	2	3	4
Er (wt %)	0	0.1	0.3	0.5
Al–Er/g	0	24.2	72	120
Al–Cu/g	0	2.7	7.6	13

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
