# Peer review of "Effect of Micro-Scale Er on the Microstructure and Fluidity of ZL205A Alloy"

_materials, 2019, doi:10.3390/ma12101688_

Round 1
Reviewer 1 Report
In this manuscript, the authors examined effect of micro-scale Er on the microstructure and fluidity of Zl205A alloy. The presented work in this manuscript is interesting, but nonetheless some changes, correction and clarifications should be introduced, in my opinion, untill publishable. I have the following comments:
1. In introduction it is indicated about high cost of technology adding Sc elements but it is not presented information concerning technology of adding Er elements. Citited literatures such as 20, 21, 22, should be presented with appropriate content from each article.
2. This paper seems interesting however it presents a lack of consistency. The analysis results are too chaotic. I suggest to introduce clear and to present crucial results of analysis. This structure of manuscript is uncomfortable to read. The Figures are placed inappropriate way what hinders reading. The Figures are not taken place near analyzed fragment.
3. Heading of chapter 2 is 'Experimental' - it is adjective. The heading may be partial.
4. The images with determining phase structures could be added.
5. The more information concerning experimental research and prepared samples should be added.
6. Some statements are inconclusive, e. g. 'Fig. 2(a) indicates that with the Er was added, the microstructure of alloy had varying degree of refinement.' - in the Figure 2a there is no added Er.
7. In the Fig. 4.: is the Fg. 4b involved analysis of EDS results in the point a from Fig. 4a and in Fig. 4c in the point b from Fig. 4a, respectively ?
8. In Fig. 7: there is duplicated 'd' symbol and there is no '7h' symbol which is specified in the article. Figures 7 (e - g) are unclear and in the Figures 7 (a - d) the scale is concealed.
9. Line 187: '7d' symbol shoul be instead of '8d'?
10. Lines 244-245: I suggest that the results shoul be from authors' analysis of results.
11. Line 282: I suggest that 'analysis results' should be instead of 'results'.
Author Response
Dear reviewers:
We thank the reviewers for reviewing our manuscript, and the referees’ reports. Based on your comment and request, we have made extensive modification on the original manuscript. Here, we attached revised manuscript. in the formats of Microsoft Word, for your approval. A document answering every question from the referees was also summarized and enclosed. A revised manuscript. with the correction sections red marked was attached as the supplemental material and for easy check/editing purpose. Should you have any questions, please contact us without hesitate.

Reviewer 2 Report
- the discussion of Er effect between line n°123 and line n°150 is no meaning because the Er was dissolved only in the matrix. The effect of Er on microstructure is not observed.
- the discussion of Er effect on T5 treatment is not explicite: the photo 7(d) has different scale thant the photos 7(a) (b) and (c), the comparaison is difficule; the EDS analysis should done for similar zone of each sample (for different precipitates and matrix); the CCC and TTT curves are only for alloys without Er addition, the use of those curves to discuss Er effect on T5 treatment is not suitable.
- in the discussion part of Er effect on fluidity, author uses argument that the Er elements are located along grain boundaries, but author has observed in fig. 5 that Er elements are located only in matrix.
- in paragraph 3.4, it is important to present mechanical properties of T5 treated samples with Er addition. In the discussion, the presence of Al8Cu4Er phase has been used as argument to justify the fracture performance but such phase has not been observed in this study.
Author Response

(The authors gave the same response as above.)

Round 2
Reviewer 2 Report
in the revised version of the manuscript, author has made necessary corrections including all remarks from reviewers.
Author Response
Dear reviewers:
We thank the reviewers for reviewing our manuscript, and the referees’ reports.